# Mental and physical health in persons receiving inpatient pulmonary rehabilitation treatment for post-COVID condition

Adrian Meule[1]*, Daniela Kroll[2,3], Martina Bönsch[3], Tessa Schneeberger[2,3], Inga Jarosch[2,3], Rainer Gloeckl[2,3], Ulrich Voderholzer[4,5,6‡], Andreas R. Koczulla[2,3,7‡]

**1** Department of Psychology, University of Regensburg, Regensburg, Germany, **2** Department of Pulmonary Rehabilitation, Philipps – University of Marburg, German Center for Lung Research (DZL), Marburg, Germany, **3** Institute for Pulmonary Rehabilitation Research, Schoen Klinik Berchtesgadener Land, Schoenau am Koenigssee, Germany, **4** Schoen Clinic Roseneck, Prien am Chiemsee, Germany, **5** Department of Psychiatry and Psychotherapy, LMU University Hospital, LMU Munich, Munich, Germany, **6** Department of Psychiatry and Psychotherapy, University Hospital of Freiburg, Freiburg, Germany, **7** Teaching Hospital, Paracelsus Medical University, Salzburg, Austria

‡ These authors share last authorship on this work.
* adrian.meule@ur.de

## Abstract

### Background

Post-COVID condition is most commonly associated with physical symptoms such as dyspnea on exertion, difficulty in concentration, fatigue, and frailty but meta-analyses also document high rates of mental health problems such as anxiety disorders, depression, and post-traumatic stress disorder (PTSD).

### Methods and findings

In the current study, 140 persons (66% female) receiving inpatient pulmonary rehabilitation treatment for post-COVID condition for an average of 27 days ($SD = 11$) completed self-report measures on mental and physical health at admission and discharge. At admission, 54%, 36%, 36%, and 14% screened positively for somatoform syndrome, generalized anxiety, depression, and PTSD, respectively. Higher pulmonary functioning related to higher self-reported physical functioning (but not to measures of mental health) at admission. Several self-reported indicators for mental and physical health improved from admission to discharge.

### Conclusions

The current study corroborates findings about the high mental and physical burden of post-COVID condition. However, both mental and physical symptoms show partial improvement during a specialized inpatient pulmonary rehabilitation treatment.

**Data availability statement:** The data and R code with which results can be reproduced are available at the Open Science Framework (https://doi.org/10.17605/OSF.IO/BYFZC).

**Funding:** The author(s) received no specific funding for this work.

**Competing interests:** The authors have declared that no competing interests exist.

## Introduction

Coronavirus disease 2019 (COVID–19) is a viral infection that is most commonly associated with fever, cough, fatigue, and dyspnea [1,2]. Post-COVID condition refers to symptoms that continue or develop after three months after a COVID–19 infection and is most commonly associated with dyspnea on exertion, difficulty in concentration, fatigue, and frailty [3,4]. Meta-analyses also document high rates of mental health problems and estimate a prevalence of approximately 25% for anxiety disorders and depression and approximately 12% for post-traumatic stress disorder (PTSD; [4–6]), possibly resulting from symptoms causing severe disruption to daily life, lack of service and treatment options, and uncertainty of illness trajectories [7]. There are several treatment options for post-COVID condition but no uniform recommendations can be made [8,9]. Thus, there is an urgent need to evaluate the effects of existing treatment approaches on mental and physical health outcomes in patients with post-COVID condition.

In the current study, we examined mental and physical health in persons receiving inpatient pulmonary rehabilitation treatment for post-COVID condition. Specifically, pulmonary functioning was assessed at admission with spirometry, blood gas analysis, and diffusion capacity for carbon monoxide, and patients completed questionnaires on mental and physical health at admission and discharge. There were three study aims: First, we examined the percentage of persons exceeding the cut-off scores across a large range of mental health problems to replicate and expand on prior findings [4–6]. Second, as it is currently unclear if and how physical and mental health impairments relate to each other in post-COVID condition, we tested whether pulmonary functioning at admission correlated with self-reported mental and physical health at admission. Third, as there is currently no "gold standard" treatment for post-COVID condition, we tested if and how self-reported mental and physical health changed during pulmonary rehabilitation treatment from admission to discharge.

## Methods

### Sample

This retrospective study of medical records using fully anonymized data did not require informed consent and was approved by the ethics committee at the University of Marburg (Marburg, Germany; reference no. 24–172 RS). Data of a consecutive sample of 140 persons who were infected with COVID–19 between 2020 and 2023 and who were admitted to inpatient pulmonary rehabilitation treatment for post-COVID condition at the Schön Klinik Berchtesgadener Land (Schönau am Königssee, Germany) between 16/01/2023 and 16/12/2023 were accessed on 19/06/2024 and analyzed. No additional inclusion or exclusion criteria were applied. Thus, time between the first COVID infection and admission to the hospital approximately ranged between 1–3 years (as the exact date of the first COVID infection could not be exactly determined retrospectively, no precise numbers can be reported here). Sample characteristics are displayed in Table 1. Treatment elements included diagnostic assessments, medical treatment, endurance and strength training, patient

**Table 1. Sample characteristics.**

| Variables | Descriptive statistics |
|---|---|
| Age (years) | *M* = 50.8, *SD* = 13.8 |
| Sex | |
| Female | *n/N* = 92/140, 65.7% |
| Male | *n/N* = 48/140, 34.3% |
| Marital status | |
| Unmarried and single | *n/N* = 15/140, 10.7% |
| Unmarried and in a relationship | *n/N* = 25/140, 17.9% |
| Married or living with partner | *n/N* = 67/140, 47.9% |
| Divorced or separated or widowed and single | *n/N* = 13/140, 9.3% |
| Divorced or separated or widowed and in a relationship | *n/N* = 20/140, 14.3% |
| Education | |
| Lower secondary education | *n/N* = 27/135, 20.0% |
| Middle secondary education | *n/N* = 32/135, 23.7% |
| Higher secondary education | *n/N* = 28/135, 20.7% |
| Tertiary education | *n/N* = 48/135, 35.6% |
| Body mass index (kg/m²) | *M* = 27.6, *SD* = 6.3 |
| Forced expiratory volume (l/s) | *M* = 2.8, *SD* = 0.9 |
| Partial pressure of oxygen at rest (mmHg) | *M* = 79.9, *SD* = 10.0 |
| Diffusing capacity of the lungs for carbon monoxide (mmol/(min*kPa)) | *M* = 7.9, *SD* = 3.7 |
| Number of times infected with COVID–19 | |
| Once | *n/N* = 133/140, 95.0% |
| Twice | *n/N* = 6/140, 4.3% |
| Three times | *n/N* = 1/140, 0.7% |
| WHO Ordinal Scale for Clinical Improvement | |
| Ambulatory: limitation of activities | *n/N* = 119/137, 86.9% |
| Hospitalized: mild disease, no oxygen therapy | *n/N* = 6/137, 4.4% |
| Hospitalized: mild disease, oxygen by mask or nasal prongs | *n/N* = 7/137, 5.1% |
| Hospitalized: severe disease, non-invasive ventilation or high-flow oxygen | *n/N* = 2/137, 1.5% |
| Hospitalized: severe disease, intubation and mechanical ventilation | *n/N* = 2/137, 1.5% |
| Hospitalized: severe disease, ventilation and additional organ support | *n/N* = 1/137, 0.7% |
| Mental disorders before first COVID–19 infection (self-reported) | |
| Depression | *n/N* = 36/140, 25.7% |
| Anxiety disorder | *n/N* = 10/140, 7.1% |
| Obsessive–compulsive disorder | *n/N* = 2/140, 1.4% |
| Post-traumatic stress disorder | *n/N* = 11/140, 7.9% |
| Eating disorder | *n/N* = 0/140, 0.0% |
| Personality disorder | *n/N* = 2/140, 1.4% |
| Schizophrenia | *n/N* = 0/140, 0.0% |
| Bipolar disorder | *n/N* = 0/140, 0.0% |
| Substance use disorder | *n/N* = 1/140, 0.7% |
| Chronic pain | *n/N* = 9/140, 6.4% |
| Other | *n/N* = 7/140, 5.0% |
| Change in mental well-being since COVID–19 infection | |
| Improved | *n/N* = 3/129, 2.3% |
| No change | *n/N* = 39/129, 30.2% |

*(Continued)*

**Table 1.** (Continued)

| Variables | Descriptive statistics |
|---|---|
| Worsened | $n/N = 65/129$, 50.4% |
| Strongly worsened | $n/N = 22/129$, 17.1% |
| Duration of inpatient pulmonary rehabilitation treatment (days) | $M = 26.9$, $SD = 11.0$ |

education, respiratory physiotherapy, relaxation techniques, occupational therapy, psychological support, and nutrition counseling [10].

## Measures

*Pulmonary functioning*. Pulmonary functioning was assessed at admission with spirometry (forced expiratory volume, $FEV_1$), blood gas analysis (partial pressure of oxygen at rest, $pO_2$), and diffusing capacity for carbon monoxide ($D_{LCO}$).

*Patient Health Questionnaire (PHQ)*. The PHQ is a self-report questionnaire for the assessment of mental disorders and associated symptoms in the past weeks [11] and patients completed the German version (PHQ–D; [12]) at admission and discharge. The PHQ includes different scales and response options and both categorical and continuous scores can be derived from item responses. Using the categorical scoring algorithms produces binary variables for the presence of somatoform syndrome (a pattern of multiple, recurrent, and frequently changing physical symptoms), depression, panic disorder, generalized anxiety, eating disorder, and alcohol use disorder. Using the continuous scoring instructions produces sum scores of items representing somatic symptoms severity (PHQ–15; for which internal consistency was $\omega = 0.81$ at admission and $\omega = 0.87$ at discharge), depressive symptoms severity (PHQ–9; for which internal consistency was $\omega = 0.84$ at admission and $\omega = 0.89$ at discharge), and stress (for which internal consistency was $\omega = 0.76$ at admission and $\omega = 0.83$ at discharge).

*Primary Care PTSD screen (PC–PTSD)*. The PC–PTSD is a self-report questionnaire for the assessment of PTSD symptoms in the past month [13] and patients completed the German version [14] at admission and discharge. It has four items with dichotomous response options (0 = no, 1 = yes). Thus, sum scores can range between 0 and 4. Internal consistency was $\omega = 0.83$ at admission and $\omega = 0.85$ at discharge. A categorical score can also be derived using a cut-off score of 3, that is, there is an indication that a person might have PTSD when at least three items are answered with yes [13].

*Short Form Health Survey (SF–36)*. The SF–36 is a self-report questionnaire for the assessment of mental and physical health in the past weeks [15,16] and patients completed the German version [17] at admission and discharge. It has 36 items with different response formats. All items are recoded such that mean scores of all subscales can range between 0 and 100, with higher values indicating better health. There are 8 subscales that represent physical functioning (for which internal consistency was $\omega = 0.93$ at admission and $\omega = 0.94$ at discharge), role limitations due to physical health (for which internal consistency was $\omega = 0.76$ at admission and $\omega = 0.81$ at discharge), role limitations due to emotional problems (for which internal consistency was $\omega = 0.88$ at admission and $\omega = 0.95$ at discharge), energy/fatigue (for which internal consistency was $\omega = 0.87$ at admission and $\omega = 0.94$ at discharge), emotional well-being (for which internal consistency was $\omega = 0.88$ at admission and $\omega = 0.91$ at discharge), social functioning (for which internal consistency was $r_{sb} = 0.84$ at admission and $r_{sb} = 0.88$ at discharge), pain (for which internal consistency was $r_{sb} = 0.91$ at admission and $r_{sb} = 0.91$ at discharge), and general health (for which internal consistency was $\omega = 0.74$ at admission and $\omega = 0.72$ at discharge).

*Other information*. Data on patients' characteristics were taken from the clinical records at the hospital (e.g., age, sex, body height and weight, number and date of COVID–19 infections and impairment according to the WHO Ordinal Scale for Clinical Improvement, date of admission and discharge at the hospital). In addition, patients completed a self-made questionnaire that included questions on marital status and education as well as a question about how mental well-being

has changed since the COVID–19 infection (response categories: strongly improved, improved, no change, worsened, strongly worsened) as part of the routine diagnostic assessment at the hospital.

## Data analyses

Data were analyzed with R version 4.3.3 in RStudio version RStudio 2024.04.1. Descriptive statistics were computed with the package *summarytools*. Internal consistencies (McDonald's ω for multi-item scales and split-half reliability with Spearman–Brown correction for two-item scales) of items used for composite scores were computed with the package *psych*. Correlations between continuous variables (Pearson's coefficient) were computed with the package *stats* and between continuous and categorical variables (point-biserial coefficients) with the package *ltm*. To test changes in self-reported mental and physical health variables from admission to discharge, linear mixed models (for continuous variables) and generalized linear mixed models (for categorical variables) that included the fixed effect of time (admission vs. discharge) and a random intercept were computed with the package *lme4*. Effect sizes (Cohen's *d* for continuous variables and Cohen's *g* for categorical variables) were computed with the package *effectsize*. Because of the numerous inferential tests and large sample size, we considered effects as significant at $p < 0.005$, as has been recommended by Benjamin and colleagues [18] who argue that this alpha level "represents 'substantial' to 'strong' evidence according to conventional Bayes factor classifications" and "would reduce the false positive rate to levels we judge to be reasonable" (p. 7). The data and R code with which results can be reproduced are available at https://osf.io/byfzc.

## Results

The majority of patients (67.5%) indicated that their mental well-being worsened or strongly worsened since the COVID–19 infection (Table 1). At admission, about half of the sample were screened positive for somatoform syndrome (54%), 36% for depression or generalized anxiety, respectively, 14% for PTSD, 9% for panic disorder, 7% for eating disorder, and none for alcohol use disorder (Table 2). Higher self-reported physical functioning related to higher forced expiratory volume and higher partial pressure of oxygen (Table 2). All other self-reported indicators of mental and physical health were unrelated to pulmonary functioning (Table 2). Somatic symptoms severity, depression, depressive symptoms severity, generalized anxiety, and stress decreased from admission to discharge with small-to-large effect sizes (Table 2). Physical functioning, energy/fatigue, and emotional well-being increased from admission to discharge with small-to-medium effect sizes (Table 2).

One of the reviewers asked whether treatment duration related to treatment outcome. Thus, we tested any *time × length of stay* interaction effects (see analysis code available here https://osf.io/byfzc), which were all $p > 0.021$, thus indicating that treatment duration did not relate to treatment outcome.

## Discussion

### Prevalence of mental health problems

In the current study, the majority of patients reported that their mental well-being worsened after the COVID–19 infection. In line with this, there were high rates of mental health problems. Specifically, 54% of the current sample screened positively for somatoform syndrome, 36% for generalized anxiety, 36% for depression, and 14% for PTSD.

The percentage of persons exceeding the somatoform syndrome cut-off score seems to be considerably higher than in other samples. For example, the prevalence of somatoform syndrome as assessed with the same instrument as in the current study (i.e., PHQ–15) was 25% in a sample of university students in Germany during the COVID–19 pandemic in 2021 [19] and, thus, less than half of the prevalence found in the current sample. Moreover, the mean somatic symptoms severity sum score was 14 in the current study and, thus, substantially higher than the score of 6 reported in sample of persons after a COVID–19 infection (with or without post-COVID condition; [20]).

**Table 2. Correlations between self-report measures of mental and physical health and pulmonary functioning indicators at admission as well as descriptive and inferential statistics for changes from admission to discharge.**

| Variables | Correlations with pulmonary functioning | | | Admission | Discharge | p | Effect size |
|---|---|---|---|---|---|---|---|
| | FEV$_1$ | pO$_2$ | D$_{LCO}$ | | | | |
| Patient Health Questionnaire | | | | | | | |
| Somatoform syndrome | −0.12 | −0.06 | −0.03 | n/N = 76/140, 54.3% | n/N = 56/134, 41.2% | 0.008 | g = 0.25 |
| Somatic symptoms severity (PHQ–15) | −0.17 | −0.01 | 0.02 | M = 13.7, SD = 5.2 | M = 12.0, SD = 5.8 | <0.001 | d = 0.45 |
| Depression | −0.15 | 0.05 | 0.04 | n/N = 50/140, 35.7% | n/N = 28/134, 20.9% | 0.002 | g = 0.29 |
| Depressive symptoms severity (PHQ–9) | −0.12 | 0.08 | 0.04 | M = 11.5, SD = 5.0 | M = 9.0, SD = 5.5 | <0.001 | d = 0.60 |
| Panic disorder | −0.19 | −0.12 | −0.18 | n/N = 13/140, 9.3% | n/N = 15/130, 11.5% | 0.241 | g = 0.12 |
| Generalized anxiety | 0.01 | −0.02 | 0.01 | n/N = 51/140, 36.4% | n/N = 35/133, 26.3% | 0.004 | g = 0.26 |
| Eating disorder | 0.04 | −0.06 | 0.09 | n/N = 10/139, 7.2% | n/N = 2/133, 1.5% | 0.010 | NA |
| Alcohol use disorder | NA | NA | NA | n/N = 0/138, 0.0% | n/N = 1/133, 0.8% | NA | NA |
| Stress | −0.03 | 0.03 | −0.01 | M = 6.9, SD = 3.8 | M = 5.7, SD = 4.0 | <0.001 | d = 0.42 |
| Primary Care PTSD Screen | | | | | | | |
| PTSD | −0.02 | 0.06 | −0.07 | n/N = 20/140, 14.3% | n/N = 18/132, 13.6% | 0.719 | g = 0.03 |
| PTSD symptoms | −0.05 | 0.10 | −0.004 | M = 0.9, SD = 1.3 | M = 0.9, SD = 1.3 | 0.879 | d = 0.01 |
| Short Form Health Survey | | | | | | | |
| Physical functioning | 0.38* | 0.27* | 0.14 | M = 47.3, SD = 24.0 | M = 57.6, SD = 25.3 | <0.001 | d = 0.56 |
| Role limitations due to physical health | 0.06 | −0.05 | −0.01 | M = 15.2, SD = 26.8 | M = 19.9, SD = 31.4 | 0.046 | d = 0.17 |
| Role limitations due to emotional problems | 0.17 | 0.12 | 0.02 | M = 53.8, SD = 44.9 | M = 57.6, SD = 47.2 | 0.403 | d = 0.06 |
| Energy/fatigue | −0.01 | −0.09 | −0.12 | M = 26.6, SD = 16.9 | M = 35.2, SD = 21.9 | <0.001 | d = 0.45 |
| Emotional well-being | 0.08 | −0.05 | 0.003 | M = 57.3, SD = 20.2 | M = 64.5, SD = 20.4 | <0.001 | d = 0.44 |
| Social functioning | 0.11 | −0.16 | −0.09 | M = 44.4, SD = 27.5 | M = 50.6, SD = 29.5 | 0.005 | d = 0.24 |
| Pain | 0.10 | 0.06 | −0.13 | M = 48.7, SD = 29.2 | M = 52.0, SD = 28.8 | 0.171 | d = 0.11 |
| General health | −0.004 | −0.21 | −0.002 | M = 36.5, SD = 16.4 | M = 40.1, SD = 16.6 | 0.006 | d = 0.24 |

*Notes.* FEV$_1$ = Forced expiratory volume, pO$_2$ = Partial pressure of oxygen, D$_{LCO}$ = Diffusing capacity of the lungs for carbon monoxide, PTSD = Post-Traumatic Stress Disorder, NA = Not available (values cannot be computed as one cell contains zero observations). Note that rules of thumb are 0.2 = small effect, 0.5 = medium effect, 0.8 = large effect for Cohen's *d* and 0.05 = small effect, 0.15 = medium effect, 0.25 = large effect for Cohen's *g* [31]. Correlations are Pearson's coefficients for continuous variables and point-biserial coefficients when one of the variables is categorical.

*$p < 0.005$.

The percentage of persons exceeding the generalized anxiety cut-off score also seems to be considerably higher than in other samples. For example, the prevalence of both panic disorder and other anxiety symptoms combined was 20% in the study by Holm-Hadulla and colleagues [19] and, thus, lower than the prevalence of generalized anxiety alone in the current sample.

The percentage of persons exceeding the depression cut-off score was not higher when compared to other samples. For example, the prevalence of depression as assessed with the same instrument as in the current study (i.e., the PHQ–9) was 42% in the study by Holm-Hadulla and colleagues [19] and, thus, slightly higher than the prevalence found in the current sample. However, the mean depressive symptoms severity sum score was 12 in the current study and, thus, higher than in other samples such as patients that presented at a post-COVID–19 clinic and in general internal medicine patients in the USA [21], a community sample of persons with or without a COVID–19 infection in the UK during the COVID–19 pandemic in 2021 [22], and patients with chronic obstructive pulmonary disease (COPD; [23]).

The percentage of persons exceeding the PTSD cut-off score was similar to a study reporting on patients recovering from COVID–19 infection in India [24]. However, when compared with studies that used the same instrument as in the current study (i.e., the PC–PTSD), it appears that the prevalence was actually lower than in other samples such as primary care patients [13], college students [25], and persons with post-COVID condition [26].

### Relationships between pulmonary functioning and self-report measures

Higher physical functioning as measured with the SF–36 related to better pulmonary functioning as measured with $FEV_1$ and $pO_2$. In contrast, pulmonary functioning was unrelated to measures of mental health in the current study. This finding is in line with studies in patients with asthma and COPD that report that higher $FEV_1$ primarily relates to higher scores on the SF–36 physical functioning subscale but not to scores on the SF–36 subscale emotional well-being [27,28]. Thus, it appears that while persons with post-COVID condition report high rates of mental health issues, these do not seem to be directly attributable to impaired pulmonary functioning.

### Changes from admission to discharge

Somatic and depressive symptoms severity, generalized anxiety, and stress as assessed with the PHQ significantly decreased from admission to discharge with small-to-large effect sizes. In contrast, symptoms of panic disorder, eating disorders, and PTSD did not change from admission to discharge, suggesting that patients with post-COVID condition presenting with these symptoms require specialized psychotherapy in addition to pulmonary rehabilitation treatment. The largest improvements on the SF–36 were observed for physical functioning, energy/fatigue, and emotional well-being in the current study. In line with this, a recent study examining effects of hyperbaric oxygenation in outpatients with post-COVID condition reported comparable SF–36 scores across subscales at the beginning of treatment and also reported the largest improvements in physical functioning, energy/fatigue, and emotional well-being [9]. As inpatient treatment is more expensive than daypatient or outpatient treatment, future studies are needed that compare different treatment elements and treatment settings in terms of cost-benefit efficacy.

### Limitations

Interpretation of results is limited to inpatients treated in Germany and may not translate to other countries with different healthcare systems. Moreover, indicators of mental and physical health were based on self-report questionnaires, which may potentially be biased (e.g., due to demand effects or social desirability, possibly leading to an overestimation of the prevalence of psychosomatic and psychological conditions or treatment effects). Finally, while this study corroborates findings about high rates of mental health problems in patients with post-COVID condition that partially improve during treatment, what this study cannot answer is the causal direction of effects. For example, it might be that mental health problems follow from the physical symptoms of post-COVID condition and resulting impairments in daily functioning [7]. However, it may also be that illness-related anxiety and dysfunctional symptom expectation contribute to persistence of physical post-COVID condition symptoms [29]. Relatedly, while preliminary findings suggest that rehabilitation interventions are superior to natural recovery [30], the current study did not include a control group of patients that were not treated or received an alternative treatment. Thus, changes from admission to discharge observed in current study cannot be causally attributed the pulmonary rehabilitation treatment.

### Conclusion

The current study dovetails with findings about the high mental and physical burden of post-COVID condition. However, both mental and physical symptoms showed partial improvement during a specialized inpatient pulmonary rehabilitation treatment. While impaired mental health may follow from physical post-COVID condition symptoms, future studies

need to address the role of pre-COVID–19 somatization tendencies in the development and maintenance of post-COVID condition.

## Author contributions

**Conceptualization:** Daniela Kroll, Martina Bönsch, Tessa Schneeberger, Inga Jarosch, Rainer Gloeckl, Ulrich Voderholzer, Andreas R. Koczulla.

**Data curation:** Adrian Meule, Daniela Kroll, Martina Bönsch, Tessa Schneeberger, Inga Jarosch, Rainer Gloeckl.

**Formal analysis:** Adrian Meule.

**Writing – original draft:** Adrian Meule.

**Writing – review & editing:** Adrian Meule, Daniela Kroll, Martina Bönsch, Tessa Schneeberger, Inga Jarosch, Rainer Gloeckl, Ulrich Voderholzer, Andreas R. Koczulla.

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
