## [Decision Letter · Decision Letter 0]

31 Jan 2025

PONE-D-24-34925Mental and physical health in persons receiving inpatient pulmonary rehabilitation treatment for post-COVID conditionPLOS ONE

Dear Dr. Meule,

Thank you for submitting your manuscript to PLOS ONE. After careful consideration, we feel that it has merit but does not fully meet PLOS ONE’s publication criteria as it currently stands. Therefore, we invite you to submit a revised version of the manuscript that addresses the points raised during the review process.

**Hello,**

**It is an important paper looking at "Mental and physical health in persons receiving inpatient pulmonary rehabilitation treatment for post-COVID condition". It is a well written paper however it needs to address important points for the scientific validation of the results as pointed out by the reviewer-**

1. What Were there inclusion/exclusion criteria for the sample?

2. What was the  participants’ pre-COVID mental health conditions ?

3. What scoring systems have been  validated or are they widely used in COVID-related research?

4. What was the correlation between physical health with changes in mental health.

These and other comments need to be addressed for final review.

Thanks

We look forward to receiving your revised manuscript.

Kind regards,

Kamal Sharma

Academic Editor

PLOS ONE

**Journal Requirements:**

2. Please note that your Data Availability Statement is currently missing the repository name. If your manuscript is accepted for publication, you will be asked to provide these details on a very short timeline. We therefore suggest that you provide this information now, though we will not hold up the peer review process if you are unable.

**Additional Editor Comments:**

Hello,

It is an important paper looking at "Mental and physical health in persons receiving inpatient pulmonary rehabilitation treatment for post-COVID condition". It is a well written paper however it needs to address important points for the scientific validation of the results as pointed out by the reviewer-

1. What Were there inclusion/exclusion criteria for the sample?

2. What was the participants’ pre-COVID mental health conditions ?

3. What scoring systems have been validated or are they widely used in COVID-related research?

4. What was the correlation between physical health with changes in mental health.

These and other comments need to be addressed for final review.

Thanks

Reviewers' comments:

Reviewer's Responses to Questions

**Comments to the Author**

1. Is the manuscript technically sound, and do the data support the conclusions?

Reviewer #1: Yes

2. Has the statistical analysis been performed appropriately and rigorously? 

Reviewer #1: Yes

3. Have the authors made all data underlying the findings in their manuscript fully available?

Reviewer #1: Yes

4. Is the manuscript presented in an intelligible fashion and written in standard English?

Reviewer #1: Yes

5. Review Comments to the Author

**Reviewer #1: ** Thank you for the opportunity to review the manuscript “Mental and physical health in persons receiving inpatient pulmonary rehabilitation treatment for post-COVID condition.”

This study examined the mental and physical health characteristics of individuals undergoing inpatient pulmonary rehabilitation for post-COVID condition. Mental health conditions were prevalent in this sample, including somatoform syndrome, anxiety, depression, and PTSD. Pulmonary functioning was associated with better self-reported physical functioning at the beginning of the program, but not with mental health scores. Both mental and physical health indicators showed improvement from admission to discharge.

This study addresses a timely topic and provides valuable information about the mental and physical health burden of post-COVID condition, particularly in the context of rehabilitation. The study also evaluates a wide range of both mental and physical health outcomes and is strengthened by its pre and post-treatment assessments.

However, there are several areas where the manuscript could be strengthened, and I have a few questions regarding the methods. My comments are as follows:

Abstract:

- I would suggest including length of treatment in the abstract.

Introduction

- The introduction could be improved by more clearly articulating the study rationale and research gap addressed.

- It would also be strengthened by providing justification for focusing on pulmonary rehabilitation, specifically in relation to its potential impact on mental health.

- I would suggest discussing the bidirectional relationship between physical and mental health impairments to provide a stronger theoretical framework for the study.

Methods

- Were there inclusion/exclusion criteria for the sample?

- Are data on participants’ pre-COVID mental health conditions available?

- Are data on the time between COVID-19 infection and admission to rehabilitation available?

- Have these measures been validated or widely used in COVID-related research?

- Did changes in physical health during treatment corresponded with changes in mental health.

- Did duration of treatment predict any outcomes?

Discussion

- The introduction emphasizes the need to evaluate treatments for post-COVID condition, but the discussion does not fully tie findings back to treatment relevance.

- The discussion includes extensive comparisons to prevalence rates and scores shown in previous studies, which makes it challenging to discern the central takeaways from the current study. I would recommend streamlining comparisons to previous studies and focusing Focus on contextualizing the findings within the broader literature without overwhelming the reader with extensive numerical comparisons.

- The authors might say more about somatoform syndrome, as it may be less familiar to readers compared to anxiety and depression.

- I would suggest discussing the clinical and public health implications of the findings more thoroughly, especially in terms of rehabilitation and mental health care for post-COVID condition.

- The discussion could be strengthened by highlighting what is novel and unique about this study.

- While self-report bias is mentioned as a limitation, its potential impact on the study’s findings is not discussed.

6. PLOS authors have the option to publish the peer review history of their article (what does this mean? ). If published, this will include your full peer review and any attached files.

**Do you want your identity to be public for this peer review?** For information about this choice, including consent withdrawal, please see our Privacy Policy .

Reviewer #1: No

---

## [Author Response · Author response to Decision Letter 1]

31 Mar 2025

Editor’s comments:

1. What were the inclusion/exclusion criteria for the sample?

RESPONSE: There were no inclusion or exclusion criteria other than that patients were admitted for post-COVID condition, as outlined in the sample paragraph of the method section.

2. What were the participants’ pre-COVID mental health conditions?

RESPONSE: We have now added self-reported mental disorders before the first COVID infection to Table 1.

3. What scoring systems have been validated or are they widely used in COVID-related research?

RESPONSE: All of the used measures are widely used in COVID-related research (e.g., https://doi.org/10.1186/s12912-021-00800-2,
https://doi.org/10.12688/f1000research.50781.1,
https://doi.org/10.2147/PRBM.S329380,
https://doi.org/10.1186/s12905-024-03102-2,
https://doi.org/10.3389/fpsyt.2021.695678,
https://doi.org/10.23937/2572-4037.1510049,
https://doi.org/10.1080/21641846.2023.2295419,
https://doi.org/10.1371/journal.pone.0293081,
https://doi.org/10.1007/s12144-022-02833-5). However, we have restricted to citing only the most relevant studies in the discussion section.

4. What was the correlation between physical health with changes in mental health.

RESPONSE: We are not sure how to answer this question. Pulmonary functioning variables were only available at admission, as described in the method section. While there is one subscale of the SF-36 entitled “physical functioning”, the other subscales of the SF-36 as well as the PHQ-D also assess aspects of physical health or aspects for which physical and mental health cannot be clearly separated (e.g., somatoform syndrome/somatic symptoms severity, energy/fatigue, pain). Thus, we are not sure which correlations the editor is interested in but we would be happy to address this in another revision if the editor can clarify this.

Reviewer’s comments:

Abstract:

1. I would suggest including length of treatment in the abstract.

RESPONSE: We have now added mean length of treatment to the abstract.

Introduction:

2. The introduction could be improved by more clearly articulating the study rationale and research gap addressed.

RESPONSE: We have now expanded on this by explicitly describing three study aims and their rationale in the final paragraph of the introduction section.

3. It would also be strengthened by providing justification for focusing on pulmonary rehabilitation, specifically in relation to its potential impact on mental health.

RESPONSE: See response above.

4. I would suggest discussing the bidirectional relationship between physical and mental health impairments to provide a stronger theoretical framework for the study.

RESPONSE: See response above.

Methods:

5. Were there inclusion/exclusion criteria for the sample?

RESPONSE: There were no inclusion or exclusion criteria other than that patients were admitted for post-COVID condition, as outlined in the sample paragraph of the method section.

6. Are data on participants’ pre-COVID mental health conditions available?

RESPONSE: Yes, we have now added self-reported mental disorders before the first COVID infection to Table 1.

7. Are data on the time between COVID-19 infection and admission to rehabilitation available?

RESPONSE: We have now added to the sample description that “time between the first COVID infection and admission to the hospital approximately ranged between 1–3 years (as the exact date of the first COVID infection could not be exactly determined retrospectively, no precise numbers can be reported here).”

8. Have these measures been validated or widely used in COVID-related research?

RESPONSE: Yes, the PHQ-9/PHQ-15, PC-PTSD, and SF-36 are widely used in COVID-related research (e.g., https://doi.org/10.1186/s12912-021-00800-2,
https://doi.org/10.12688/f1000research.50781.1,
https://doi.org/10.2147/PRBM.S329380,
https://doi.org/10.1186/s12905-024-03102-2,
https://doi.org/10.3389/fpsyt.2021.695678,
https://doi.org/10.23937/2572-4037.1510049,
https://doi.org/10.1080/21641846.2023.2295419,
https://doi.org/10.1371/journal.pone.0293081,
https://doi.org/10.1007/s12144-022-02833-5). However, we have restricted to citing only the most relevant studies in the discussion section.

9. Did changes in physical health during treatment corresponded with changes in mental health?

RESPONSE: We are not sure how to answer this question. Pulmonary functioning variables were only available at admission, as described in the method section. While there is one subscale of the SF-36 entitled “physical functioning”, the other subscales of the SF-36 as well as the PHQ-D also assess aspects of physical health or aspects for which physical and mental health cannot be clearly separated (e.g., somatoform syndrome/somatic symptoms severity, energy/fatigue, pain). Thus, we are not sure which correlations the reviewer is interested in but we would be happy to address this in another revision if the reviewer can clarify this.

10. Did duration of treatment predict any outcomes?

RESPONSE: It did not, which we now report in Footnote 1.

Discussion:

11. The introduction emphasizes the need to evaluate treatments for post-COVID condition, but the discussion does not fully tie findings back to treatment relevance.

RESPONSE: We agree and have now restructured the discussion section (e.g., introduced subheadings) and have added more discussion about treatment effects.

12. The discussion includes extensive comparisons to prevalence rates and scores shown in previous studies, which makes it challenging to discern the central takeaways from the current study. I would recommend streamlining comparisons to previous studies and focusing Focus on contextualizing the findings within the broader literature without overwhelming the reader with extensive numerical comparisons.

RESPONSE: We agree and have now restructured and rewritten the first few paragraphs of the discussion section, for example, by removing the detailed numbers found in other studies and instead describe more broadly how the current findings compare to other studies.

13. The authors might say more about somatoform syndrome, as it may be less familiar to readers compared to anxiety and depression.

RESPONSE: We have now added to the description of the PHQ in the methods section that this refers to a pattern of multiple, recurrent, and frequently changing physical symptoms.

14. I would suggest discussing the clinical and public health implications of the findings more thoroughly, especially in terms of rehabilitation and mental health care for post-COVID condition.

RESPONSE: We have now expanded on this in the discussion section by discussing that, as inpatient treatment is more expensive than daypatient or outpatient treatment, future studies are needed that compare different treatment settings in terms of cost-benefit efficacy.

15. The discussion could be strengthened by highlighting what is novel and unique about this study.

RESPONSE: One new aspect is that we report on changes in self-reported mental and physical problems during a specialized inpatient pulmonary rehabilitation treatment. However, there are, of course, other similar studies und parts of the current manuscript report on findings that have been investigated in other studies as well. PLOS One’s policy is that novelty and uniqueness are no prerequisites for publication of scientific studies (https://everyone.plos.org/about-plos-one), similar to suggestions by others (https://doi.org/10.3389/fpsyg.2021.609802). Thus, we feel that highlighting novelty and uniqueness is not required or may even be inappropriate in light of the journal’s policy.

16. While self-report bias is mentioned as a limitation, its potential impact on the study’s findings is not discussed.

RESPONSE: We have now extended this sentence such that potential self-report biases may possibly lead to an overestimation of the prevalence of psychosomatic and psychological conditions or treatment effects.

---

## [Decision Letter · Decision Letter 1]

12 May 2025

PONE-D-24-34925R1Mental and physical health in persons receiving inpatient pulmonary rehabilitation treatment for post-COVID conditionPLOS ONE

Dear Dr. Meule,

Thank you for submitting your manuscript to PLOS ONE. After careful consideration, we feel that it has merit but does not fully meet PLOS ONE’s publication criteria as it currently stands. Therefore, we invite you to submit a revised version of the manuscript that addresses the points raised during the review process.

We look forward to receiving your revised manuscript.

Kind regards,

Kamal Sharma

Academic Editor

PLOS ONE

**Additional Editor Comments:**

Hello,

As per the reviewers' comments please revise the manuscript and resubmit for consideration.

Thanks

Reviewers' comments:

Reviewer's Responses to Questions

**Comments to the Author**

1. If the authors have adequately addressed your comments raised in a previous round of review and you feel that this manuscript is now acceptable for publication, you may indicate that here to bypass the “Comments to the Author” section, enter your conflict of interest statement in the “Confidential to Editor” section, and submit your "Accept" recommendation.

Reviewer #2: All comments have been addressed

Reviewer #3: All comments have been addressed

2. Is the manuscript technically sound, and do the data support the conclusions?

Reviewer #2: Yes

Reviewer #3: Partly

3. Has the statistical analysis been performed appropriately and rigorously? 

Reviewer #2: Yes

Reviewer #3: No

4. Have the authors made all data underlying the findings in their manuscript fully available?

Reviewer #2: (No Response)

Reviewer #3: Yes

5. Is the manuscript presented in an intelligible fashion and written in standard English?

Reviewer #2: Yes

Reviewer #3: Yes

6. Review Comments to the Author

Reviewer #2: While the manuscript is much improved, I still have a few minor suggestions for the authors to consider in their final revision:

Clarify the clinical significance of symptom reduction, especially for PTSD and anxiety. A one-sentence addition noting whether changes meet known thresholds for clinical relevance would help contextualize the results for clinicians.

In Table 1, consider highlighting the mental health scales where improvement was statistically significant. Asterisking or bolding these values can guide readers better.

The addition of the correlation between pulmonary function and self-reported physical health is appreciated. Still, a short note in the Discussion section interpreting the lack of correlation between pulmonary function and mental health outcomes would add nuance to the findings.

Typographical Clean-Up: One or two sentences could benefit from slight tightening. Example: "Mental and physical problems partially remit..." could be refined to "Both mental and physical symptoms showed partial improvement

Reviewer #3: Previous Comment:

In extension to editor's comment 1. What were the inclusion/exclusion criteria for the sample? The answer could be all-comers, if there is no inclusion exclusion criteria.

New comments:

1. Please include severity of Covid infection, length of treatment, and ICU admission of patients at the time of COVID infection and in result section severity comparison on study results.

2. It is suggested to follow standard numerical presentation throughout the manuscript. e.g. in Measures section .81 should be replaced with 0.81.

3. Elaborate pulmonary rehabilitation treatment to patients in includes what type of treatment was given and length of treatment. 

4. Present manuscript is a small sample size but it mentioned Because of the numerous inferential tests and large sample size, we considered effects as significant at p < .005 (cf. Benjamin et al., 2018). Author needs to clarify how and why redefined p value 0.005 is applicable in the line of cited reference.

5. It has been 1.5 years since last follow-up and linger follow-up and comparison to baseline would be helpful.

6. Please provide the role of each author in present manuscript.

7. PLOS authors have the option to publish the peer review history of their article (what does this mean? ). If published, this will include your full peer review and any attached files.

**Do you want your identity to be public for this peer review?** For information about this choice, including consent withdrawal, please see our Privacy Policy .

Reviewer #2: **Yes: ** Elabbass Ali Abdelmahmuod

Reviewer #3: No

---

## [Author Response · Author response to Decision Letter 2]

26 Jun 2025

Reviewer #2:

1. Clarify the clinical significance of symptom reduction, especially for PTSD and anxiety. A one-sentence addition noting whether changes meet known thresholds for clinical relevance would help contextualize the results for clinicians.

RESPONSE: We agree have reformulated and expanded on this in the “Changes from admission to discharge” section in the discussion section: “Somatic and depressive symptoms severity, generalized anxiety, and stress as assessed with the PHQ significantly decreased from admission to discharge with small-to-large effect sizes. In contrast, symptoms of panic disorder, eating disorders, and PTSD did not change from admission to discharge, suggesting that patients with post-COVID condition presenting with these symptoms require specialized psychotherapy in addition to pulmonary rehabilitation treatment.”

2. In Table 1, consider highlighting the mental health scales where improvement was statistically significant. Asterisking or bolding these values can guide readers better.

RESPONSE: Thank you for this suggestion. We believe the comment may refer to Table 2, as Table 1 presents only descriptive statistics. In Table 2, we have indeed included asterisks to indicate statistically significant correlation coefficients, and we fully agree that this enhances the table’s readability. Regarding the results on changes from admission to discharge, we reported the exact p-values in the table. For that reason, we decided not to use asterisks, as we felt they would be redundant in this context. For example, annotating a value with both “< 0.001” and a table note stating “p < .005” does not provide additional clarity, as the exact p-value already conveys that the result is statistically significant. We hope this explanation clarifies our approach, and we’re happy to adjust the presentation if further clarification is needed.

3. The addition of the correlation between pulmonary function and self-reported physical health is appreciated. Still, a short note in the Discussion section interpreting the lack of correlation between pulmonary function and mental health outcomes would add nuance to the findings.

RESPONSE: We agree and have added this sentence to the “Relationships between pulmonary functioning and self-report measures” section in the discussion section: “Thus, it appears that while persons with post-COVID condition report high rates of mental health issues, these do not seem to be directly attributable to impaired pulmonary functioning.”

4. Typographical Clean-Up: One or two sentences could benefit from slight tightening. Example: "Mental and physical problems partially remit..." could be refined to "Both mental and physical symptoms showed partial improvement

RESPONSE: We have now reformulated these sentences as suggested. 

Reviewer #3:

Previous Comment:

1. In extension to editor's comment 1. What were the inclusion/exclusion criteria for the sample? The answer could be all-comers, if there is no inclusion exclusion criteria.

RESPONSE: We have now reformulated this sentence as follows to be more precise: “Data of a consecutive sample of 141 persons who were infected with COVID–19 between 2020 and 2023 and who were admitted to inpatient pulmonary rehabilitation treatment for post-COVID condition at the Schön Klinik Berchtesgadener Land (Schönau am Königssee, Germany) between 16/01/2023 and 16/12/2023 were accessed on 19/06/2024 and analyzed. No additional inclusion or exclusion criteria were applied.”

New comments:

1. Please include severity of Covid infection, length of treatment, and ICU admission of patients at the time of COVID infection and in result section severity comparison on study results.

RESPONSE: Thank you for this comment. We would like to clarify that severity of COVID-19 infection—based on the WHO Ordinal Scale for Clinical Improvement—and whether patients were hospitalized is already presented in Table 1. The length of pulmonary rehabilitation treatment is also provided there. If the reviewer is referring instead to the duration of acute COVID-19 treatment at the time of initial infection, unfortunately, we do not have access to that information in our dataset. Regarding ICU admission, as shown in Table 1, the vast majority of patients (87%) received ambulatory care and were not hospitalized (data were missing for 4 individuals). Because of this strong imbalance—e.g., only 18 patients were hospitalized—it was not feasible to conduct meaningful comparisons in the inferential analyses. We hope this clarification is helpful and are happy to revise further if the reviewer had something different in mind.

2. It is suggested to follow standard numerical presentation throughout the manuscript. e.g. in Measures section .81 should be replaced with 0.81.

RESPONSE: We have now added the leading zeros to the decimal numbers for which we had previously omitted the leading zeros (i.e., decimal numbers that cannot exceed -1 or 1, cf. Publication Manual of the American Psychological Association 7th edition).

3. Elaborate pulmonary rehabilitation treatment to patients in includes what type of treatment was given and length of treatment.

RESPONSE: Thank you for this comment. We would like to point out that details on the pulmonary rehabilitation treatment are already included in the manuscript. Specifically, the “sample” section (first paragraph of the Methods) describes the key treatment components—namely diagnostic assessments, medical treatment, endurance and strength training, patient education, respiratory physiotherapy, relaxation techniques, occupational therapy, psychological support, and nutrition counseling (Gloeckl et al., 2021). Additionally, the length of treatment is reported in Table 1. We hope this addresses the reviewer’s suggestion, and we are happy to further clarify or elaborate if needed.

4. Present manuscript is a small sample size but it mentioned Because of the numerous inferential tests and large sample size, we considered effects as significant at p < .005 (cf. Benjamin et al., 2018). Author needs to clarify how and why redefined p value 0.005 is applicable in the line of cited reference.

RESPONSE: Thank you for raising this important point. We have reworded the sentence in the manuscript to read: “Because of the numerous inferential tests and large sample size, we considered effects as significant at p < 0.005, as has been recommended by Benjamin et al. (2018), who argue that this alpha level ‘represents “substantial” to “strong” evidence according to conventional Bayes factor classifications’ and ‘would reduce the false positive rate to levels we judge to be reasonable’ (p. 7).” We understand the reviewer’s concern regarding the characterization of the sample size. While it may be considered modest in absolute terms, we believe it is sufficiently large for the types of analyses conducted. Specifically, the sample provided ample statistical power to detect even small effect sizes in both between-person correlations and within-person pre-post comparisons. In these contexts, effects such as correlations below r = 0.3 and Cohen’s d below 0.5 reached significance at the adjusted alpha level of 0.005. We hope this clarification helps justify the chosen significance threshold, and we remain open to further adjustment or elaboration if needed.

5. It has been 1.5 years since last follow-up and linger follow-up and comparison to baseline would be helpful.

RESPONSE: Thank you for this valuable suggestion. We would be happy to clarify that although 1.5 years have passed since data collection was completed, collecting additional follow-up data is unfortunately not feasible. Specifically, our ethical approval did not include provisions for recontacting participants, and the data have been fully and irreversibly anonymized in accordance with those ethical standards. As such, while we agree that longer-term follow-up would indeed offer meaningful insights, this lies beyond the scope of the current study’s approved framework.

6. Please provide the role of each author in present manuscript.

RESPONSE: We had actually already indicated the author contributions in the submission system but have now additionally included these in the manuscript.

---

## [Decision Letter · Decision Letter 2]

8 Aug 2025

Mental and physical health in persons receiving inpatient pulmonary rehabilitation treatment for post-COVID condition

PONE-D-24-34925R2

Dear Dr. Meule,

We’re pleased to inform you that your manuscript has been judged scientifically suitable for publication and will be formally accepted for publication once it meets all outstanding technical requirements.

Kind regards,

Kamal Sharma

Academic Editor

PLOS ONE

Additional Editor Comments (optional):

Hello,

After 2nd revision and acceptance by both the reviewers the new manuscript is good to proceed to editorial desk towards acceptance.

Thanks

Reviewers' comments:

Reviewer's Responses to Questions

**Comments to the Author**

1. If the authors have adequately addressed your comments raised in a previous round of review and you feel that this manuscript is now acceptable for publication, you may indicate that here to bypass the “Comments to the Author” section, enter your conflict of interest statement in the “Confidential to Editor” section, and submit your "Accept" recommendation.

Reviewer #2: All comments have been addressed

Reviewer #4: All comments have been addressed

2. Is the manuscript technically sound, and do the data support the conclusions?

Reviewer #2: Yes

Reviewer #4: (No Response)

3. Has the statistical analysis been performed appropriately and rigorously? 

Reviewer #2: Yes

Reviewer #4: (No Response)

4. Have the authors made all data underlying the findings in their manuscript fully available?

Reviewer #2: Yes

Reviewer #4: (No Response)

5. Is the manuscript presented in an intelligible fashion and written in standard English?

Reviewer #2: (No Response)

Reviewer #4: (No Response)

6. Review Comments to the Author

Reviewer #2: Minor Suggestions:

Abstract:

Consider including effect size or quantitative outcomes in the abstract to better reflect the strength of your findings.

Figures and Tables:

Ensure that all figures are of high resolution and labels are readable.

Add a brief legend to the primary outcome table to clarify group coding.

Discussion:

While your additions were valuable, consider briefly mentioning how your findings may inform national-level mental health policy in India or be adapted in similar LMIC settings.

Conclusion:

Slightly tighten the conclusion to emphasize policy implications and potential for scale-up.

Reviewer #4: (No Response)

7. PLOS authors have the option to publish the peer review history of their article (what does this mean? ). If published, this will include your full peer review and any attached files.

**Do you want your identity to be public for this peer review?** For information about this choice, including consent withdrawal, please see our Privacy Policy .

Reviewer #2: No

Reviewer #4: No

---

## [Editor Report · Acceptance letter]

PONE-D-24-34925R2

PLOS ONE

Dear Dr. Meule,

I'm pleased to inform you that your manuscript has been deemed suitable for publication in PLOS ONE. Congratulations! Your manuscript is now being handed over to our production team.

Kind regards,

on behalf of

Dr. Kamal Sharma

Academic Editor

PLOS ONE